# The Acute Effects of Caffeine Supplementation on Anaerobic Performance and Functional Strength in Female Soccer Players

**DOI:** 10.3390/nu17132156

**Published:** 2025-06-28

**Authors:** Hakkı Mor, Ahmet Mor, Mekki Abdioğlu, Dragoș Ioan Tohănean, Cătălin Vasile Savu, Gizem Ceylan Acar, Cristina Elena Moraru, Dan Iulian Alexe

**Affiliations:** 1Department of Coaching Education, Yaşar Doğu Faculty of Sport Sciences, Ondokuz Mayıs University, Samsun 55200, Turkey; hakki.mor@omu.edu.tr; 2Department of Coaching Education, Faculty of Sport Sciences, Sinop University, Sinop 57010, Turkey; amor@sinop.edu.tr; 3Graduate School of Health Sciences, Ankara University, Ankara 06110, Turkey; mekkiabdioglu@gmail.com; 4Department of Motor Performance, Transilvania University of Brașov, Eroilor nr. 29, 500036 Brașov, Romania; 5Department of Sport Games and Physical Education, “Dunărea de Jos” University of Galati, 800008 Galați, Romania; catalin.savu@ugal.ro; 6Department of Sports Management, Faculty of Sport Sciences, Muş Alparslan University, Muş 49250, Turkey; g.ceylan@alparslan.edu.tr; 7Department of Physical Education and Sport, “Alexandru Ioan Cuza” University of Iasi, 700506 Iaşi, Romania; 8Department of Physical and Occupational Therapy, “Vasile Alecsandri” University of Bacau, 600115 Bacau, Romania; alexedaniulian@ub.ro

**Keywords:** caffeine, anaerobic performance, female soccer players, functional strength, ball speed

## Abstract

**Background/Objectives:** Despite extensive research on caffeine’s (CAF’s) ergogenic effects, evidence regarding its impact on anaerobic performance in female athletes remains limited and inconclusive. The aim of this study was to investigate the acute effects of 6 mg/kg^−1^ caffeine on anaerobic performance, functional strength, agility, and ball speed in female soccer players. **Methods:** A randomized, double-blind, placebo-controlled crossover design was employed. Thirteen moderately trained female soccer players (age: 21.08 ± 1.11 years; height: 161.69 ± 6.30 cm; weight: 59.69 ± 10.52 kg; body mass index (BMI): 22.77 ± 3.50 kg/m^2^; training age: 7.77 ± 1.16 years; habitual caffeine intake: 319 ± 160 mg/day) completed two experimental trials (caffeine vs. placebo (PLA)), separated by at least 48 h. Testing sessions included performance assessments in vertical jump (VJ), running-based anaerobic sprint test (RAST), bilateral leg strength (LS), handgrip strength (HS), single hop for distance (SH), medial rotation (90°) hop for distance (MRH), change of direction (COD), and ball speed. Rating of perceived exertion (RPE) was also recorded. **Results:** CAF ingestion significantly improved minimum (*p* = 0.011; d = 0.35) and average power (*p* = 0.007; d = 0.29) during RAST. A significant increase was also observed in SHR (single leg hop for distance right) performance (*p* = 0.045; d = 0.44). No significant differences were found in VJ, COD, ball speed, LS, HS, SHL, MRHR, or MRHL (*p* > 0.05). RPE showed a moderate effect size (d = 0.65) favoring the CAF condition, though not statistically significant (*p* = 0.110). **Conclusions:** In conclusion, acute CAF intake at a dose of 6 mg/kg^−1^ may enhance anaerobic capacity and lower-limb functional strength in female soccer players, with no significant effects on jump height, agility, or upper-body strength.

## 1. Introduction

Caffeine’s removal from the World Anti-Doping Agency’s (WADA’s) list of banned substances in 2004 has sparked an exponential rise in scientific research into CAF’s potential as an ergogenic aid [1]. Initially, CAF was believed to increase exercise performance metabolically [2]. However, lately, the most promising ergogenic feature has been suggested to be associated with CAF’s action on the central nervous system (CNS) through adenosine receptor blockage (A1 and A2A) resulting in increased neuronal excitability which stimulates alertness, decreases perceived effort, and eventually improves performance after binding to these receptors [3]. Another mechanism by which CAF exerts its effect on skeletal muscle is an increase in Na++/K++ pump activation [4]. Lastly, CAF also raises blood norepinephrine levels and glycolytic activity and affects the neuromuscular system, increasing motor unit recruitment and contractile force [5].

Numerous studies have been conducted on the effects of CAF on endurance exercise, concluding that CAF improves performance in long-term aerobic-type activities [6]. Hodgson et al. [7] observed performance times during cycling trials were significantly faster (5%) for 5 mg/kg CAF compared to PLA. CAF has also been shown to increase endurance performance in reviews and meta-analyses conducted over the previous two decades [6,8]. On the other hand, fewer studies have been conducted on the effects of CAF on explosive-type activities, with inconsistent findings. Ellis et al. [9] concluded that acute CAF intake resulted in a significant increase in agility, 20 m sprint, and countermovement jump (CMJ) performance in male soccer players. Additionally, acute CAF improved lower and upper body strength performance, even if the ergogenic magnitude was negligible [10]. In contrast, there are studies which concluded that CAF has no effect on these parameters [11,12,13]. The differences in these results might be attributed to participants’ training status and sex, as well as various testing procedures.

Males and females may respond differently to CAF, including physiological changes and performance consequences, as a result of variations in body size, lean body mass, and systemic steroid hormone levels [14]. Additionally, studies on the impact of CAF on female individuals’ anaerobic performance, such as agility, repeated sprint running, and strength, are likewise scarce and conflicting [15]. In fact, several experts have emphasized the need to investigate the effects of CAF on female performance [10,16,17]. Due to the ambiguous results in the literature so far, Grgic and Del Coso [10] suggested that future research assess lower body exercise performance in females with CAF supplementation. In a recent review by Mielgo-Ayuso et al. [18], CAF supplementation was suggested to produce a similar ergogenic aid for aerobic but not for anaerobic exercise performance in males and females. Since just 13% of the participants in the literature examining the ergogenic effects of CAF were female, generalizing the findings to females would be challenging. Of these investigations with females, refs. [16,19,20] were hampered by the use of tests with low external validity (e.g., isokinetic dynamometry and cycle ergometers, etc.), as well as the inclusion of females who were using oral contraceptives, which have been demonstrated to interact with caffeine metabolism [2].

The recent literature underscores the paucity and inconsistency of evidence regarding the ergogenic effects of caffeine (CAF) in female athletes. Addressing this gap, Bougrine et al. [21] demonstrated that a morning intake of 6 mg/kg CAF yielded significantly greater improvements in anaerobic performance compared to lower doses. Complementarily, Siquier-Coll et al. [22] observed the beneficial effects of a 5 mg/kg CAF supplementation protocol on both physical performance and perceived fatigue during a one-week training cycle in female volleyball players. Furthermore, Bougrine et al. [23] reported that combining CAF ingestion with motivational music resulted in superior anaerobic performance outcomes compared to either strategy applied independently. Supporting these findings, another study by Bougrine et al. [24] comparing CAF doses of 3, 6, and 9 mg/kg identified 6 mg/kg^−1^ as the most efficacious dose for enhancing short-term maximal performance in young female team-sport athletes.

Although aerobic metabolism supplies a football game’s essential physiological and metabolic requirements, high-level abilities such as jumping, kicking, and tackling are anaerobic [25]. Ball speed can be by far the most crucial ability of a football player, which was shown by Rada et al. [26] as a new efficient performance indicator. Only two studies have assessed the effects of CAF on ball speed. López-Samanes et al. [27] examined the impact of a 3 mg/kg dose in male futsal players and reported no ergogenic benefits. On the other hand, Mor et al. [28] investigated the effects of a 6 mg/kg CAF dose in moderately trained male football players and reported a 2.7% increase in ball speed performance, although this improvement did not reach statistical significance. Therefore, the aim of the present study, for the first time, was to investigate the effects of acute 6 mg/kg^−1^ of CAF on vertical jumping, running-based anaerobic sprint test (RAST), lower body strength, agility, and ball speed performance in a single field test battery in female football players.

## 2. Materials and Methods

### 2.1. Participants

Thirteen healthy, non-smoking, young, moderately-trained [29], female football players participated in this study (age 21.08 ± 1.11 years, height 161.69 ± 6.30 cm, weight 59.69 ± 10.52 kg, BMI 22.77 ± 3.50 kg/m^2^ habitual consumption of CAF: 319 ± 160 mg/day^−1^; mean ± SD). The required sample size was determined using G*Power software (Heinrich Heine University Düsseldorf, version 3.1.9.2, Düsseldorf, Germany). Based on an effect size of 0.50, a confidence interval of 1 − β = 0.95, a significance level of α = 0.05, and an actual power of 0.96, it was calculated that a total of eight participants would be sufficient for the study [28]. All participants had an average of 7.77 ± 1.16 years of experience competing at the club level in football and at least 2 ± 2 years of participation in regional and university competitions. Participants were eligible for inclusion if they were non-smokers aged between 18 and 40 years, had no history of cardiopulmonary disease, and had not experienced any musculoskeletal injuries within the three months preceding the study. Additional inclusion criteria required engagement in regular football training, defined as a minimum of five sessions per week (approximately 7.5 h), and the absence of any use of ergogenic substances known to affect hydration status or exercise performance during the prior three months. Individuals were excluded if they had used any medication within the past month, had a known allergy or hypersensitivity to caffeine, or were currently using oral contraceptives. Prior to their inclusion in the study, all participants were thoroughly informed about the experimental procedures, after which they provided their written informed consent. The study was conducted in full compliance with the ethical principles outlined in the Declaration of Helsinki and received approval from the Human Research Ethics Committee at Sinop University (Reference number: E-57452775-044-92026, date 24 March 2022). The study’s flow diagram and schematic representation of the study design are illustrated in Figure 1 and Figure 2, respectively.

### 2.2. Experimental Design

During the initial laboratory visit, participants underwent anthropometric and body composition assessments (e.g., body mass and stature) and completed a habitual caffeine consumption questionnaire. Habitual CAF intake was evaluated using a validated instrument [30]. Only individuals reporting a daily CAF intake below 350 mg·d^−1^ were included in the study to control for individual variability in CAF responsiveness due to habituation. On this first visit, participants also performed the testing protocol at a low intensity for familiarization, ensuring that no substantial exertion was induced.

Following the familiarization session, participants were randomly assigned in a double-blind, crossover, counterbalanced design to ingest either CAF or PLA. A minimum washout period of 48 h was observed between testing sessions to ensure complete caffeine clearance and sufficient recovery. Participants were instructed to document their dietary intake during the 24 h preceding the first testing session and to replicate the same dietary pattern before the subsequent session.

To control for potential confounding factors, participants were instructed to avoid the intake of CAF, ergogenic aids (such as nitrate and sodium bicarbonate), alcohol, and anti-inflammatory medications. Moreover, they were advised to limit any high-intensity exercise during the 24 h preceding each experimental session. Additionally, they were advised to maintain consistency in nutrition and sleep.

Considering circadian rhythm effects, all testing and measurements were conducted at the same time of day (between 5:00 and 7:00 p.m.). On the testing days, outdoor environmental conditions were recorded as follows: ambient temperature ranged between 8–9 °C, humidity between 73–81%, and wind speed between 7.5–9.3 km/h. To ensure the reliability of the study, all tests and measurements were performed under controlled indoor conditions with an ambient temperature of 20–24 °C at Sinop University’s indoor sports hall and performance measurement laboratory. To ensure a double-blind design, the preparation of both caffeine and placebo capsules and the random assignment of trial order was conducted by a researcher who was not involved in the data collection process. The randomization was performed using specialized software (www.randomizer.org). The same researcher was also responsible for distributing the capsules to participants prior to each trial and verifying their ingestion [31]. Participants were required to wear identical clothing and footwear across all test sessions. Each session began with a standardized 15 min warm-up (arm circles and crossovers, forward–backward and lateral leg swings, dynamic hip openers, walking lunges, straight-leg march (toy soldiers), high knees, butt kicks, side shuffles, carioca (grapevine), quick-feet drills, skipping with arm swings, bounding strides, and progressively accelerating short sprints), and ad libitum water consumption was permitted throughout both trials.

The experimental testing protocol in each session included the following sequence of performance assessments: functional performance tests (single leg hop for distance and 90° medial rotation hop tests), vertical jump, leg strength, handgrip strength, change-of-direction (COD) performance via the Illinois agility test, ball speed, and anaerobic capacity via the Running-based Anaerobic Sprint Test (RAST). These tests were selected for their relevance to the movement patterns observed in soccer training and competition. To facilitate recovery, a passive rest period of 3 min was provided between each performance test, excluding the vertical jump test. At the conclusion of each testing session, participants’ perceived exertion was measured using the Borg Rating of Perceived Exertion (RPE) scale (0–10) [32].

#### 2.2.1. Procedures

##### Anthropometric and Body Composition Assessments

All anthropometric and body composition assessments were performed during the initial familiarization session. Body mass (kg) was measured using a bioelectrical impedance analysis device (BIA, InBody 120, InBody Co., Ltd., Seoul, Korea), while height (cm) was assessed with a portable stadiometer (Seca 213, Hamburg, Germany). Subsequently, participants’ body mass index (BMI) was calculated by dividing their body mass by the square of their height in meters (kg/m^2^).

##### Running-Based Anaerobic Sprint Test (RAST)

The Running-based Anaerobic Sprint Test (RAST) was employed to evaluate anaerobic capacity. In brief, the RAST comprised six repeated 35 m maximal sprints, each separated by a 10 s recovery period (inclusive of the deceleration phase). An infrared photocell device (Seven, SE-165 Photocell Stopwatch, Istanbul, Turkey) with a precision of ±0.01 s, positioned exactly 35 m apart, was utilized to record the time required for each sprint interval. Prior to the test, athletes’ body weights were measured. Athletes were instructed to perform a 15 min warm-up to ensure both mental and physical readiness. During the RAST, each participant performed six consecutive 35 m sprints, each followed by a 10-s rest interval. After the initial sprint and subsequent 10 s recovery period, the timing device emitted a beep signaling the start of the next sprint. This procedure was repeated until all six sprints were completed in the same manner. For data collection and analysis, an assistant recorded the time for each 35 m sprint to the nearest hundredth of a second and carried out the necessary calculations. The power output for each sprint was determined using the following formulas: Velocity = Distance ÷ Time; Acceleration = Velocity ÷ Time; Force = Weight × Acceleration; and Power = Force × Velocity or alternatively, Power = Weight × Distance^2^ ÷ Time^3^ [28].

##### Vertical Jump and Anaerobic Power Test

The vertical jump performance of the participants was assessed using a digital vertical jump device (Takei 5406 Jump-MD Vertical Jumpmeter, Tokyo, Japan). Initially, the rubber jump plate was positioned on a level surface. To control for the potential influence of different footwear, all participants were instructed to remove their shoes and assume a “ready” stance with bare feet centered on the plate, keeping a distance of 10–20 cm between their feet. Following this, the researcher, who was consistently responsible for fastening the digital belt to ensure test reliability, reset the digital belt to zero, securely fastened it around the participant’s waist, and gently rotated the pulley in the direction indicated by the arrow to remove any slack from the rope. Once the participant was prepared, they rapidly transitioned from an upright standing posture to a 90° knee flexion with a free arm swing, executing a maximal vertical jump. The trial was repeated if the participant stepped forward during the jump, the measuring tape became loose, or if they failed to land back on the rubber plate. Each participant completed two trials, with a 1 min rest interval between jumps. The highest jump was recorded in centimeters, with a precision of ±1 cm [33]. Participants’ anaerobic power calculations were executed using the Lewis formula: Anaerobic Power (W) = {√ 4.9 [Body Weight (kg)] √ Vertical Jump (m)}.

##### Illinois Agility Test

Agility performance was evaluated using the change of Direction (COD) test, measured using a photocell timing system (Seven, SE-165 Photocell Stopwatch, Istanbul, Turkey) with a precision of ±0.01 s. The agility course consisted of a 10 m long and 5 m wide area, in which four 12-inch traffic training cones were positioned in a straight line at 3.3 m intervals along the center line. The test protocol included a 40 m linear sprint and a 20 m slalom run involving 180° directional changes every 10 m. Photocell timing gates were positioned at both the starting and finishing lines at an approximate height of 1 m. Upon readiness, participants commenced the test from a position 30 cm behind the start line and were instructed to perform at maximal effort. Each athlete completed two trials, interspersed with 3 min passive recovery periods, and the best performance time was recorded as the Illinois Agility Test score [34].

##### Ball Speed Test

Ball speed was assessed from a distance of 11 m (penalty mark) from the goal using a radar gun (Bushnell Velocity Speed Gun, Overland Park, KS, USA), capable of measuring speeds within a range of 16–177 km/h with an accuracy of ±2 km/h. Prior to testing, participants’ dominant legs were identified through subjective self-report to ensure maximal kicking performance. Subsequently, participants executed shots employing the instep kick technique. All kicks were performed using a size 5 soccer ball (appropriate for individuals aged 12 years and older) in accordance with FIFA regulations. To maintain consistency and enhance reliability, the same researcher conducted all measurements by positioning the radar gun behind the goal, directly aligned with the penalty spot from which the ball was kicked. Participants were instructed to aim for accuracy (targeting) while simultaneously achieving maximal ball speed. Each player was given two trials to obtain the best score, and the results were recorded in km/h [28].

##### Leg Strength Test

The subjects’ leg strengths were determined using a back-leg dynamometer (Takei TKK-5402, Tokyo, Japan) with accuracy up to 300 kg. The device allows measurement based on differences in subjects’ height or center of applied force using an adjustable-length chain. During dynamometry, participants positioned their feet on the dynamometer and maintained their arms at their sides, with their backs slightly bending forward. Participants were instructed to pull the dynamometer bar with their hands at maximum velocity, relying on just their legs and not their backs, until their knees were fully extended, which required no more than 3 s, as previously described. The test was performed two times with a three-minute rest between each trial, and the best score was recorded [35].

##### Handgrip Strength Test

A hand grip dynamometer (Takei TKK 5401, Tokyo, Japan) was used to measure grip strength. The dynamometer was set to ensure that the participant could comfortably grip the handle with the palmar eminences and intermediate phalanges. The subject stood still with the arm to be measured straight and without touching the body, and the measurement was taken when the arm was at a 45° angle to the body. The test was performed twice for the right and left hands, and the best score was recorded [36].

##### Functional Performance Tests

The single leg hop test (SLHT) protocols that the researchers had previously established were implemented in this investigation. The protocol previously described by Bishop et al. [37] was used to conduct the single hop for distance (SH) test. We adhered to the protocol meticulously defined by Dingenen et al. [38] for the medial rotation (90°) hop for distance (MRH).

A cloth tape, which was affixed to the ground in a perpendicular position to a beginning line, was 6 m in length and 5 cm in width. There was a 30 cm starting location for the tape. The participants conducted each functional test by stepping out from behind the starting line. While waiting on the hopping limb behind the starting line, the participants executed their maximal hop with arm swing. The trial was repeated after a 60 s rest if the subject was unable to maintain the final landing for a minimum of 2 s or land on the other limb. The evaluations were administered in the following order, starting with the right limb and ending with the left limb. Each SLHT was administered to the participants twice, with a three-minute passive rest interspersed between each test. The researcher measured the distance from the starting line to the precise position of the heel landing on the ground, and the best trial was recorded in centimeters for analysis.

Participants were instructed to hop as far forward as possible along the tape measure line and land on the same limb for the SH. In MRH, the medial side of the foot was required to remain perpendicular to the hop direction while the participants stood on the limb to be tested. During the swing phase, the individual executed a single leg hop in the transversal plane, rotating 90° medially. Before takeoff, the foot was instructed to avoid rotating in the direction of the hop. The foot should be positioned in the direction of the tape measure line on landing [37,38].

##### Supplementation Protocol

Participants ingested 6 mg/kg of caffeine (Nature’s Supreme, Istanbul, Turkey) or placebo supplements (wheat bran) encapsulated in gelatin hard capsules of the same color and form, administered 60 min prior to the testing protocol [2]. The CAF dose was prepared at room temperature by a researcher who was otherwise not involved in the study, using electronic laboratory scales with a sensitivity of one milligram.

### 2.3. Statistical Analyses

Data were checked for normality by using the Shapiro–Wilk test. Between-group comparisons were analyzed with a paired sample t-test in functional performance tests, vertical jump, leg strength, handgrip strength, agility, ball-kicking speed, and anaerobic capacity performances. Effect size was determined using Cohen’s d (large: d > 0.8; moderate: d = 0.8 to 0.5; small: d = 0.5 to 0.2; trivial: d < 0.2) [39]. Statistical significance was set at *p* < 0.05. All data were analyzed using SPSS version 27.0 (IBM Corp., Armonk, NY, USA) and are presented as mean ± standard deviation (SD).

## 3. Results

Participants had a mean age of 21.08 ± 1.11 years, with an average body mass of 59.69 ± 10.52 kg and height of 161.69 ± 6.30 cm. The mean body mass index (BMI) was 22.77 ± 3.50 kg/m^2^. The group’s mean training experience was 7.77 ± 1.16 years, and their habitual daily CAF intake averaged 319 ± 160 mg (Table 1).

RAST results are presented in Table 2. Minimum power output was significantly higher in the CAF condition compared to PLA (179.32 ± 67.32 vs. 156.97 ± 60.15 W; *p* = 0.011; d = 0.35). Similarly, average power output increased following CAF ingestion (241.55 ± 73.84 vs. 221.30 ± 64.15 W; *p* = 0.007; d = 0.29). Maximum power output also showed an increase under CAF (309.32 ± 77.64 vs. 291.00 ± 61.06 W), though the difference did not reach statistical significance (*p* = 0.076; d = 0.26). No statistically significant differences were observed between conditions for vertical jump height (VJ, cm), vertical jump power (VJ, W), fatigue index (W/s), change-of-direction time (s), or ball speed (km/h), with all comparisons yielding *p* > 0.05 and effect sizes ranging from trivial to small (d ≤ 0.16).

Strength and functional lower-limb performance results are shown in Table 3. A statistically significant difference was detected only in SH_R_ (*p* = 0.045; d = 0.44). No significant between-group differences were observed for HS_R_ (*p* = 0.260; d = 0.26), HS_L_ (*p* = 0.253; d = 0.28), LS (*p* = 0.083; d = 0.48), SH_L_ (*p* = 0.248; d = 0.34), MRH_R_ (*p* = 0.901; d = 0.01), or MRH_L_ (*p* = 0.062; d = 0.43).

The rating of perceived exertion (RPE) did not differ significantly between conditions (*p* = 0.110); however, the CAF trial yielded a lower mean value (2.92 ± 0.64) than PLA (3.46 ± 0.97), with a moderate effect size (d = 0.65) (Figure 3).

## 4. Discussion

The aim of this study is to investigate the effects of acute caffeine consumption on anaerobic capacity, jumping, strength, agility, and ball speed in female football players. To the best of our knowledge, this study is the first to examine the influence of acute caffeine consumption on specific performance parameters. The main finding of this study is that the intake of 6 mg/kg caffeine enhances anaerobic capacity and lower extremity functional strength. In addition, a moderate effect size in favor of the caffeine group was observed in RPE. No difference was found between PLA and caffeine in terms of jump, strength, agility, and ball speed.

The current study showed that the consumption of 6 mg/kg of CAF increased anaerobic capacity in female football players. However, previous studies have stated that acute caffeine intake does not affect anaerobic performance in female athletes [40,41]. In contrast, several studies, in line with our findings, have concluded that moderate doses of acute CAF consumption improve anaerobic capacity [42,43]. The current findings show partial alignment with those of Bougrine et al. [24], who reported that a 6 mg/kg dose of caffeine enhanced repeated sprint and agility performance. In our study, comparable significant improvements were observed in RAST scores. Nevertheless, the lack of significant changes in agility and vertical jump outcomes may reflect variability arising from task specificity or differences in sample characteristics. The effects of CAF on anaerobic performance can be explained by both peripheral and central mechanisms. Specifically, at the peripheral level, CAF supplementation may enhance neuromuscular functioning and increase the bioavailability of calcium in the myoplasm, which may improve muscular power output. Meanwhile, at the central level, CAF acts as an adenosine antagonist, increasing neurotransmitter synthesis and stimulating the nervous system [42]. Therefore, to determine the optimal dosage for achieving the maximum effect of CAF, further research should be conducted to examine and verify the changes in these mechanisms of action and to investigate the neuromuscular responses to CAF supplementation during anaerobic tasks in football players. Importantly, although CAF is widely recognized as an effective ergogenic aid, the majority of the evidence supporting its use is derived from studies conducted predominantly with male participants [18,44,45]. Nonetheless, studies specifically addressing female athletes are emerging. Similarly, Lara et al. [43] investigated whether the ergogenic effect of CAF on anaerobic performance was comparable between male and female athletes. The researchers examined the impact of a 3 mg/kg acute CAF dose using the Wingate test and found, consistent with our findings, that acute CAF consumption improved both peak and mean power in both sexes. Nonetheless, although previous research suggests that CAF’s effects on performance are similar between males and females, there remains a need for studies that include female participants [43,46]. Accordingly, Clarke et al. [46] demonstrated that consuming coffee containing 3 mg/kg of CAF elevated salivary CAF levels and improved 5 km cycling time trial performance similarly in both males and females. Furthermore, Goldstein et al. [47] suggested that an acute intake of 6 mg/kg CAF may enhance strength performance in females with resistance training experience. However, no significant changes were found in muscular endurance. Additionally, Karayiğit et al. [48] investigated the separate and combined effects of CAF (6 mg/kg) and taurine (1 g) on female team athletes, administered 60 min before exercise. A PLA group received 300 mg of maltodextrin. Their results showed that the combined intake significantly improved both peak and mean power, whereas neither supplement alone produced similar effects. Our findings support previous research and provide further evidence for the ergogenic potential of CAF supplementation in enhancing anaerobic performance.

Moreover, several studies have demonstrated that CAF, when consumed as a dietary supplement, can influence various components of exercise performance in individuals who are regular CAF users. It has been suggested that CAF doses ranging between 3 and 6 mg/kg, consumed approximately 60 min prior to exercise, may be sufficient to enhance performance [15,45,49,50]. Nevertheless, despite our study’s observed increase in lower extremity functional strength, no improvements were found in vertical jump, leg strength, hand grip strength, agility, or ball speed. This may be due to the training status of the athletes, as previous research has suggested that CAF’s ergogenic effects are more prominent in highly trained individuals [51]. Thus, the relatively moderate training background of our participants may explain the lack of response compared to elite or professional athletes assessed in earlier studies [52]. Bougrine et al. [21] identified a more pronounced ergogenic response to caffeine when administered in the morning hours, suggesting a potential chronobiological influence on its effectiveness. Given that the present study was conducted in the evening, this temporal factor may partially account for the discrepancies observed. Furthermore, Bougrine et al. [23] highlighted that combining caffeine with motivational music may yield synergistic benefits beyond those achieved by caffeine alone. In light of this, future research should investigate integrated protocols involving caffeine intake and music exposure within football contexts. Specifically, performance in jumping tasks depends on mechanical efficiency during concentric and eccentric phases, elastic element contribution, and nervous system characteristics. Caffeine ingestion has been shown to enhance muscle activation and excitation–contraction coupling during high-intensity, rapid movements. These mechanisms may influence various kinematic and kinetic parameters of jumping, potentially resulting in improved performance. Ultimately, caffeine remains one of the most widely used ergogenic aids among athletes in sports where jumping ability is crucial. However, its performance-enhancing effects appear to be independent of an athlete’s training level [53]. For instance, López-Samanes et al. [27] examined the effects of acute CAF on performance in their study with futsal players. The authors reported that acute CAF consumption significantly improved CMJ and sprint performance, although the observed increase in ball speed did not reach statistical significance. Conversely, Fernández-Campos et al. [40] found that the consumption of caffeine-containing energy drinks did not improve physical performance parameters, such as grip strength, jumping ability, or anaerobic capacity, in professional female volleyball players. However, findings from another study involving female athletes showed that a 6 mg/kg dose of acute CAF improved jump performance, although no significant gains were observed in muscular strength [54]. Surprisingly, Norum et al. [55] demonstrated that a 4 mg/kg dose of CAF increased maximal strength, power, and muscular endurance in resistance-trained and caffeine-habituated female athletes, thereby supporting its use as an effective ergogenic aid both before competition and during training.

Given the limited number of studies examining the impact of CAF on ball speed parameters, further research is needed to thoroughly investigate, evaluate, and conceptualize its potential effects in this area. In the present study, CAF supplementation was found to enhance performance in the single-leg hop for distance and medial rotation hop tests, both of which assess lower extremity functional strength. This improvement may be attributed to increased calcium release from the sarcoplasmic reticulum and enhanced force production. Additionally, the observed performance gains could be partially explained by CAF’s stimulatory effect on central nervous system excitability, potentially enhancing motor unit recruitment [56]. In support of these findings, Ali et al. [20] investigated the effects of 6 mg/kg CAF intake on knee flexor and extensor strength in female athletes and reported that CAF improved eccentric strength and power both during an intermittent running protocol and in a subsequent assessment conducted the following morning. Similarly, Merino Fernández et al. [57] demonstrated that a 3 mg/kg CAF dose led to significant improvements in both bilateral and unilateral vertical jump performance among elite Jiu-Jitsu athletes, thereby supporting the recommendation of CAF as an effective ergogenic aid. In line with these results, Pérez-López et al. [58] examined various physical performance parameters—including grip strength, jumping ability, and agility—in female athletes who consumed either an energy drink containing 3 mg/kg of CAF or a PLA. The findings indicated that the caffeinated beverage significantly enhanced physical performance compared to the PLA condition. Furthermore, Lara et al. [43] evaluated the effects of different supplementation protocols and concluded that an energy drink containing 3 mg/kg of CAF, when ingested 60 min prior to exercise, effectively improved performance variables such as jump height, peak power, and maximal speed during CMJ and average peak running speed in 7 × 30 m sprint test in female football players. Importantly, the study also reported that this supplementation did not lead to an increased incidence of adverse effects following the exercise session [43]. In contrast to these findings, our study observed no improvement in leg strength values following CAF consumption. Consistently, Jones et al. [59] reported that CAF doses of 3 and 6 mg/kg did not affect lower body maximal strength but did enhance lower body muscular endurance in female with resistance training experience. Additionally, Filip-Stachnik et al. [60] found that, among female habituated to CAF, acute intake of 3 and 6 mg/kg CAF increased maximal strength at both doses, while only minimal ergogenic effects were observed on strength-endurance performance. Moreover, their findings suggest that CAF at a dose of 6 mg/kg may yield a greater performance benefit compared to the lower dose.

The removal of caffeine from the World Anti-Doping Agency’s list of prohibited substances in 2004, combined with a growing body of scientific evidence supporting its ergogenic potential, has contributed to a notable rise in CAF use among both male and female athletes across various sports disciplines in recent years [15,44]. One of the key explanations for this trend is that CAF is believed to exert its performance-enhancing effects on the central nervous system by antagonizing adenosine receptors, a mechanism that may lead to pain attenuation and, consequently, a reduction in rating of perceived exertion (RPE) [2]. In the present study, although no statistically significant differences were observed in RPE scores between the groups, a moderate effect size in favor of the CAF group was detected, suggesting a potential practical benefit. Similarly, Bougrine et al. [61], in their research with female athletes, reported that CAF intake had no significant impact on RPE. In contrast, Salgueiro et al. [62] presented inconsistent findings, indicating that CAF consumption may reduce RPE values in athletes.

Although the findings of this study provide valuable contributions, certain limitations must be acknowledged. Firstly, the participant group consisted of regional-level female football players with moderate training backgrounds, which limits the generalizability of the results to elite-level athletes. Secondly, the application of multiple physical performance tests within a single session may have led to cumulative fatigue, potentially affecting the outcomes of subsequent tests. Thirdly, the lack of control over the participants’ menstrual cycle phases may have resulted in the oversight of possible hormonal influences on performance. Fourthly, the study employed only a single CAF dose (6 mg/kg) and, therefore, did not assess a dose–response relationship. Lastly, no nutritional survey was conducted in the study. Additionally, central nervous system-related performance parameters (e.g., cognitive reaction time, attention level) were not evaluated. These factors constrain a more comprehensive understanding of the effects of CAF supplementation. The results of this study offer practical implications for coaches and practitioners aiming to enhance athletic performance through informed supplementation strategies. Acute supplementation with a 6 mg/kg dose of CAF was found to be effective, especially in enhancing anaerobic capacity and lower extremity functional strength. In this context, coaches might consider implementing CAF supplementation in a controlled and individualized manner prior to training sessions or matches requiring high-intensity exertion. Furthermore, despite the observed improvements in certain parameters, the absence of significant changes in vertical jump, agility, and ball speed suggests that the effects of CAF may vary depending on the type of movement. Moreover, the moderate effect size observed in favor of CAF regarding RPE indicates that it may help reduce subjective fatigue and enhance motivational levels. While the small effect size observed in RPE may seem statistically modest, it could hold practical relevance by contributing to sustained motivation and training adherence in female team sports. In applied settings, such incremental improvements may positively influence the consistency of performance across training phases or competitive scenarios. In a comparable context, Siquier-Coll et al. [22] demonstrated that caffeine supplementation positively influenced both physical performance and perceived fatigue in female volleyball players. These findings underscore caffeine’s potential as a practical ergogenic aid for female team sport athletes under both training and competitive conditions. Particularly in female athletes exposed to high training loads, CAF supplementation could serve as a performance-enhancing strategy, provided that individual tolerance levels are taken into account during dosage adjustments.

## 5. Conclusions

This study may contribute to a better understanding of the ergogenic effects of acute CAF intake and its potential to enhance performance in female athletes. However, based on the findings, it can be stated that the ingestion of 6 mg/kg CAF 60 min prior to training had no significant effect on vertical jump, agility, ball speed, or handgrip strength in female football players. On the other hand, the results indicate that CAF supplementation improved anaerobic performance and functional strength in the lower extremity. Taken together, these findings suggest that acute CAF intake may serve as an effective ergogenic aid for enhancing certain aspects of performance in female athletes. Future research should more comprehensively evaluate the effects of CAF supplementation on performance in female athletes by controlling for biological variables such as hormonal fluctuations. Additionally, studies comparing different CAF dosages and the timing of ingestion would contribute to optimizing the efficacy of this supplement. Moreover, investigations supported by additional physiological measures, such as cognitive performance, blood lactate concentration, and anaerobic capacity, may provide a more detailed understanding of CAF’s central and peripheral effects. Finally, systematically monitoring the duration and severity of gastrointestinal symptoms would offer valuable insights regarding the safety of CAF use.

## Figures and Tables

**Figure 1 nutrients-17-02156-f001:**
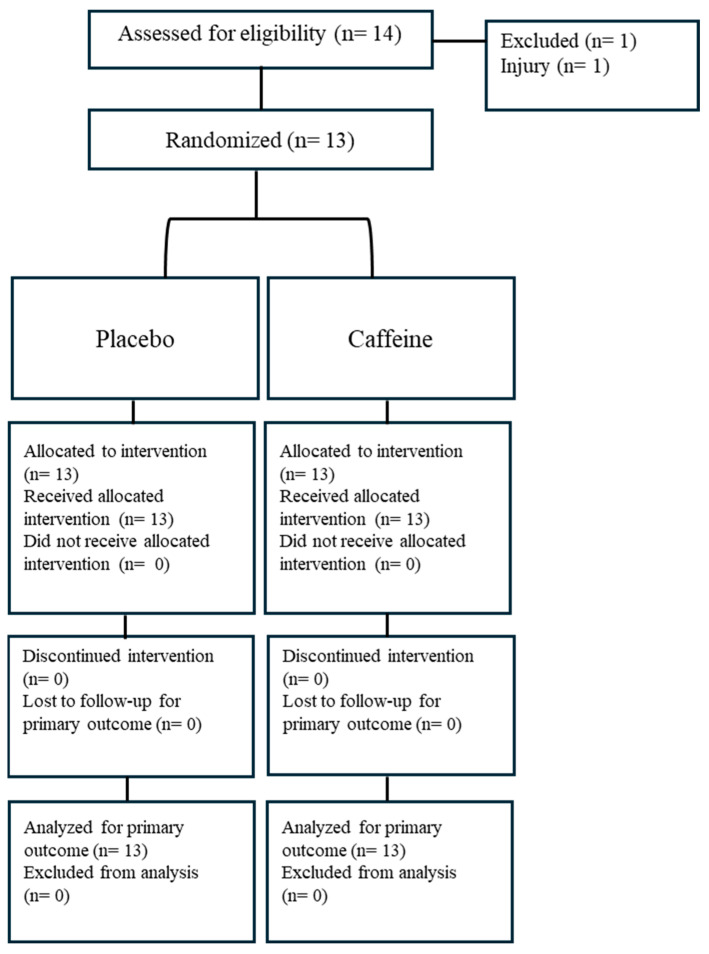
Consort flow diagram.

**Figure 2 nutrients-17-02156-f002:**
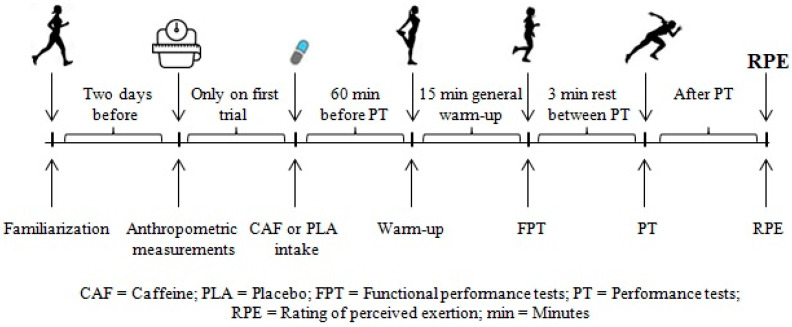
A schematic illustration of the experimental design.

**Figure 3 nutrients-17-02156-f003:**
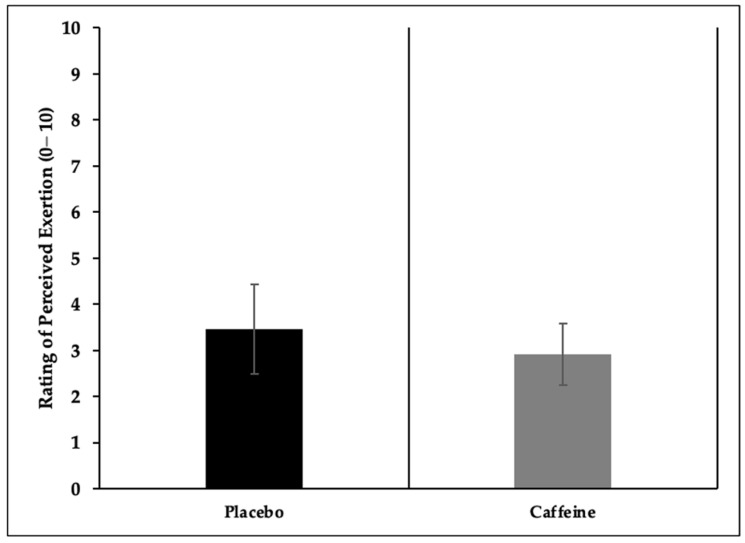
Rating of perceived exertion.

**Table 1 nutrients-17-02156-t001:** Descriptive statistics (n = 13).

Variables	X	SD
Age (yr)	21.08	1.11
Height (cm)	161.69	6.30
Weight (kg)	59.69	10.52
BMI (kg/m^2^)	22.77	3.50
Training age (yr)	7.77	1.16
Habitual consumption of CAF (mg/day^−1^)	319	160

X = Mean; SD = Standard deviation.

**Table 2 nutrients-17-02156-t002:** Differences in the mean values between the CAF and PLA groups in PT.

	Groups			
Variables	PLA	CAF	d	*p*
	X ± SD	X ± SD		
VJ (cm)	44.54 ± 7.54	43.69 ± 5.87	*0.12*	0.540
VJ (WATT)	879.33 ± 168.89	873.90 ± 173.70	*0.03*	0.728
Maximum power (W)	291.00 ± 61.06	309.32 ± 77.64	*0.26*	0.076
Minimum power (W)	156.97 ± 60.15	179.32 ± 67.32	** *0.35* **	**0.011 ***
Average power (W)	221.30 ± 64.15	241.55 ± 73.84	** *0.29* **	**0.007 ***
Fatigue index (W/s)	3.15 ± 0.61	3.16 ± 0.88	*0.01*	0.959
COD (s)	18.43 ± 0.76	18.48 ± 0.85	*0.06*	0.724
Ball Speed (km/s)	66.47 ± 9.11	67.84 ± 8.12	*0.16*	0.341

* (*p* < 0.05); X = Mean; SD = Standard deviation; d = Cohen’s d effect size; VJ = Vertical jump; COD = Change of direction (Illinois agility test).

**Table 3 nutrients-17-02156-t003:** Differences in the mean values between the CAF and PLA groups in FPT.

	Groups			
Variables	PLA	CAF	d	*p*
	X ± SD	X ± SD		
HS_R_ (kg)	28.52 ± 4.28	27.57 ± 4.11	*0.22*	0.176
HS_L_ (kg)	25.75 ± 5.01	25.15 ± 4.66	*0.12*	0.403
LS (kg)	72.61 ± 21.96	72.35 ± 22.10	*0.11*	0.901
SH_R_ (cm)	124.80 ± 15.72	132.39 ± 18.06	*0.44*	**0.045 ***
SH_L_ (cm)	125.30 ± 14.40	129.02 ± 20.31	*0.21*	0.234
MRH_R_ (cm)	110.99 ± 14.56	118.48 ± 16.38	*0.48*	0.062
MRH_L_ (cm)	108.26 ± 12.74	113.09 ± 17.09	*0.32*	0.170

* (*p* < 0.05); X = Mean; SD = Standard deviation; d = Cohen’s d effect size; HS_R_ = Handgrip strength right; HS_L_ = Handgrip strength left; LS = Leg Strength; SH_R_ = Single leg hop for distance right; SH_L_ = Single leg hop for distance left; MRH_R_ = 90^o^ Medial rotation hop test right; MRH_L_ = 90° Medial rotation hop test left.

## Data Availability

The authors confirm the data supporting the findings of this study are available within the article. Raw data that support the findings of this study are available from the corresponding author, upon reasonable request.

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
