# Peer review of "The Acute Effects of Caffeine Supplementation on Anaerobic Performance and Functional Strength in Female Soccer Players"

_nutrients, 2025, doi:10.3390/nu17132156_

Round 1
Reviewer 1 Report
Comments and Suggestions for Authors
Introduction
The introduction is well constructed and provides a good background. However, in the justification, you point out the scarcity of literature on the effect of caffeine on women. Please consult the recent literature:
DOI: 10.3390/nu16010029
DOI: 10.3390/nu17101613
DOI: 10.3390/nu16142223
DOI: 10.3390/nu16050640
Participants
Please indicate which previous study you based your sample calculation on and cite it.
Please indicate the menstrual phase of the participants.
Provide more details on the inclusion and exclusion criteria and add the CONSORT flow diagram.
Was a nutritional survey conducted?
Add more information on how the warm-up was conducted.
Provide more information on how the researchers were blinded and how randomisation was performed.
Discussion
Please compare with the current scientific literature.
DOI: 10.3390/nu16010029
DOI: 10.3390/nu17101613
DOI: 10.3390/nu16142223
DOI: 10.3390/nu16050640
Author Response
Dear Reviewer 1
We know that time is an important resource and we sincerely thank you for taking the time to help us in our endeavor to promote our idea.
Response to Reviewer 1
Dear Reviewer, we would like to thank you for taking the time to meticulously review our manuscript and for your insightful comments and suggestions, which have significantly improved the quality of this manuscript. We have responded to each comment in a point-by-point fashion below.
Introduction
The introduction is well constructed and provides a good background. However, in the justification, you point out the scarcity of literature on the effect of caffeine on women. Please consult the recent literature:
DOI: 10.3390/nu16010029
DOI: 10.3390/nu17101613
DOI: 10.3390/nu16142223
DOI: 10.3390/nu16050640
A paragraph containing the existing literature has been added to the introduction section and highlighted in lines 89-100 in the manuscript.
Participants
Please indicate which previous study you based your sample calculation on and cite it.
Revised and highlighted in line 122 in the manuscript.
Please indicate the menstrual phase of the participants.
Information regarding participants' menstrual status was not obtained, and this has been explicitly acknowledged as a limitation in lines 502-504 in the manuscript.
Provide more details on the inclusion and exclusion criteria and add the CONSORT flow diagram.
Revised and highlighted in lines 124-132 in the manuscript and a consort flow diagram was added.
Was a nutritional survey conducted?
No nutritional survey was conducted.
Add more information on how the warm-up was conducted.
Revised and highlighted in lines 178-182 in the manuscript.
Provide more information on how the researchers were blinded and how randomisation was performed.
Revised and highlighted in lines 171-177 in the manuscript.
Discussion
Please compare with the current scientific literature.
DOI: 10.3390/nu16010029
DOI: 10.3390/nu17101613
DOI: 10.3390/nu16142223
DOI: 10.3390/nu16050640
The existing literature has been added to the discussion section and highlighted in lines 376-381, lines 422-429, and lines 522-526 in the manuscript.

Reviewer 2 Report
Comments and Suggestions for Authors
Dear authors,
Congratulations on your well-structured and methodologically sound study investigating the acute effects of caffeine supplementation on anaerobic performance and functional strength in female soccer players. The manuscript demonstrates scientific rigor and offers meaningful contributions to the field of sports nutrition and performance enhancement in female athletes. Below is a section-by-section evaluation of the manuscript, along with a few minor suggestions to further refine and perfect your submission.
ABSTRACT
The abstract is well-structured and scientifically grounded, providing adequate information regarding the study's purpose, design, and main findings. It effectively summarizes the key elements of the research.
INTRODUCTION
The introduction presents a strong and well-documented scientific rationale for the study, clearly outlining both the theoretical and applied significance of the research. It establishes a solid foundation for the investigation.
MATERIALS AND METHODS
This section is methodologically sound, comprehensive, and well-organized. It meets the essential criteria for a high-quality experimental study and supports the internal validity of the research.
However, while the study mentions "random assignment," the exact method of randomization (e.g., use of software, sealed envelopes, random number generator) is not specified. Including this detail would enhance transparency and methodological clarity. Additionally, the subtitle “Procedures” (line 168) appears unnumbered, which could be adjusted for consistency.
RESULTS
The results section is statistically thorough and clearly presented. However, it would be beneficial to explicitly highlight instances where statistical significance is achieved but the effect size remains small, to help readers better interpret the practical relevance of the findings.
DISCUSSION
The discussion is rich in references and successfully integrates the current findings with prior literature. Introducing subsections could improve the readability and structure of this extensive section. Furthermore, greater emphasis on interpreting small effect sizes, beyond their statistical significance, would strengthen the discussion's depth and practical insights.
CONCLUSIONS
The conclusion is appropriately balanced and scientifically sound, aligning well with the reported findings. It provides a concise and careful summary, suitable for a scientific article.
Overall, this is a valuable and timely study. The aforementioned refinements are offered to support the continuous improvement of an already commendable manuscript. Well done!
Author Response
Dear Reviewer 2
We know that time is an important resource and we sincerely thank you for taking the time to help us in our endeavor to promote our idea.
Dear Reviewer, we would like to thank you for taking the time to meticulously review our manuscript and for your insightful comments and suggestions, which have significantly improved the quality of this manuscript. We have responded to each comment in a point-by-point fashion below.
Dear authors,
Congratulations on your well-structured and methodologically sound study investigating the acute effects of caffeine supplementation on anaerobic performance and functional strength in female soccer players. The manuscript demonstrates scientific rigor and offers meaningful contributions to the field of sports nutrition and performance enhancement in female athletes. Below is a section-by-section evaluation of the manuscript, along with a few minor suggestions to further refine and perfect your submission.
Abstract
The abstract is well-structured and scientifically grounded, providing adequate information regarding the study's purpose, design, and main findings. It effectively summarizes the key elements of the research.
Thank you for your remarks.
Introduction
The introduction presents a strong and well-documented scientific rationale for the study, clearly outlining both the theoretical and applied significance of the research. It establishes a solid foundation for the investigation.
Thank you for your remarks.
Materials and Methods
This section is methodologically sound, comprehensive, and well-organized. It meets the essential criteria for a high-quality experimental study and supports the internal validity of the research.
However, while the study mentions "random assignment," the exact method of randomization (e.g., use of software, sealed envelopes, random number generator) is not specified. Including this detail would enhance transparency and methodological clarity. Additionally, the subtitle “Procedures” (line 168) appears unnumbered, which could be adjusted for consistency.
We appreciate your insightful feedback. The subtitle ‘Procedures’ was numbered and the list was revised. The randomization method was also added and highlighted in lines 171-177 in the manuscript.
Results
The results section is statistically thorough and clearly presented. However, it would be beneficial to explicitly highlight instances where statistical significance is achieved but the effect size remains small, to help readers better interpret the practical relevance of the findings.
Revised and highlighted in the manuscript.
Discussion
The discussion is rich in references and successfully integrates the current findings with prior literature. Introducing subsections could improve the readability and structure of this extensive section. Furthermore, greater emphasis on interpreting small effect sizes, beyond their statistical significance, would strengthen the discussion's depth and practical insights.
Revised and highlighted in lines 518-522 in the manuscript.
Conclusions
The conclusion is appropriately balanced and scientifically sound, aligning well with the reported findings. It provides a concise and careful summary, suitable for a scientific article.
Overall, this is a valuable and timely study. The aforementioned refinements are offered to support the continuous improvement of an already commendable manuscript. Well done!
Thank you for your remarks.

Reviewer 3 Report
Comments and Suggestions for Authors
Please see my comments and recommendations. The comment on the results needed to be improved refers only to the habitual intake of caffeine.
First, I would like to recognize the authors for the work they have submitted.
The abstract is well-presented and includes all the necessary information, including the purpose of the study, the methodology, main results (including effect sizes) and the general conclusion.
The introduction is right to the point and presents the relevant literature to support the variables of interest. Also, it includes studies using a 3-5 mg/kg dose of caffeine, which may support the selection of 6 mg/kg selected by the investigators.
Methods
-8 participants may be considered enough to find differences, but considering the inter-individual variation in caffeine metabolism, I consider the sample size small.
- I am a bit confused, as you have stated that your participants had a habitual consumption of caffeine 319 +/- 160 mg/day initially, while later on (line 129), you mention that only individuals reporting a daily CAF below 250mg/day were included. Which one is true?
-If you have a study that can support the order of your testing, that would be great, as this is an intense testing to be done on the same day. I am not saying that it cannot be done, but if you have a study that had the same testing in the past, that will support your methodology.
-The in-text citation for Cohen (1998) should be presented as a numbered citation in brackets, in accordance with the journal’s guidelines. It should appear as reference number [34].
Results
-Please review the mean value of CAF that you included in Table 1 and revise your inclusion criteria accordingly.
The rest of the findings are well-presented and supported.
Discussion
Why did you choose the 6 mg/kg dosage? I understand that you have presented a study on male football players where 6 mg/kg was used. However, that study specifically involved male football players. Is there a particular reason for selecting 6 mg/kg instead of 5 or 4 mg/kg?. A similar study demonstrated improvements in certain parameters even with the 3mg/kg.
The rest of the discussion is well presented and supported.
In addition to the small sample size, I consider the lack of control for menstrual cycle phase and oral contraceptive use to be significant limitations, considering the study was done only in female subjects. Hormonal fluctuations throughout the menstrual cycle can include a range of physiological responses, including strength, endurance, thermoregulation and substrate metabolism.
Despite the limitations, I still consider the findings important to publish.
Author Response
Dear Reviewer 3
Please see the attachment.
We know that time is an important resource and we sincerely thank you for taking the time to help us in our endeavor to promote our idea.

Round 2
Reviewer 1 Report
Comments and Suggestions for Authors
The authors have responded appropriately to my comments. Please indicate in the limitations that no nutritional survey was conducted.
Reviewer 3 Report
Comments and Suggestions for Authors
I am happy with the changes and revisions!